# As^(III, V)^ Uptake from Nanostructured Iron Oxides and Oxyhydroxides: The Complex Interplay between Sorbent Surface Chemistry and Arsenic Equilibria

**DOI:** 10.3390/nano12030326

**Published:** 2022-01-20

**Authors:** Marco Sanna Angotzi, Valentina Mameli, Alessandra Fantasia, Claudio Cara, Fausto Secci, Stefano Enzo, Marianna Gerina, Carla Cannas

**Affiliations:** 1Department of Chemical and Geological Sciences, University of Cagliari, S.S. 554 bivio per Sestu, 09042 Monserrato, Italy; marcosanna@unica.it (M.S.A.); fantasia.91@hotmai.it (A.F.); cara.claudio16@gmail.com (C.C.); fausto.s1993@gmail.com (F.S.); ccannas@unica.it (C.C.); 2Consorzio Interuniversitario Nazionale per la Scienza e Tecnologia dei Materiali (INSTM), Via Giuseppe Giusti 9, 50121 Firenze, Italy; 3Department of Chemistry and Pharmacy, University of Sassari, Via Vienna 2, 07100 Sassari, Italy; enzo@uniss.it; 4Department of Inorganic Chemistry, Charles University, Hlavova 8, 12800 Prague, Czech Republic; marygerina@hotmail.it

**Keywords:** akaganeite, ferrihydrite, maghemite, arsenic, water remediation, β-FeOOH

## Abstract

Iron oxides/oxyhydroxides, namely maghemite, iron oxide-silica composite, akaganeite, and ferrihydrite, are studied for As^V^ and As^III^ removal from water in the pH range 2–8. All sorbents were characterized for their structural, morphological, textural, and surface charge properties. The same experimental conditions for the batch tests permitted a direct comparison among the sorbents, particularly between the oxyhydroxides, known to be among the most promising As-removers but hardly compared in the literature. The tests revealed akaganeite to perform better in the whole pH range for As^V^ (max 89 mg g^−1^ at pH_0_ 3) but to be also efficient toward As^III^ (max 91 mg g^−1^ at pH_0_ 3–8), for which the best sorbent was ferrihydrite (max 144 mg g^−1^ at pH_0_ 8). Moreover, the study of the sorbents’ surface chemistry under contact with arsenic and arsenic-free solutions allowed the understanding of its role in the arsenic uptake through electrophoretic light scattering and pH measurements. Indeed, the sorbent’s ability to modify the starting pH was a crucial step in determining the removal of performances. The As^V^ initial concentration, contact time, ionic strength, and presence of competitors were also studied for akaganeite, the most promising remover, at pH_0_ 3 and 8 to deepen the uptake mechanism.

## 1. Introduction

Arsenic pollution in surface and groundwater is a worldwide issue due to its natural abundance by dissolution from soils or anthropogenic activities [1]. The arsenic toxicity depends on its chemical nature, as inorganic arsenic compounds are more dangerous than organic ones. Moreover, factors, such as pH, redox potential, competitors, and microorganisms may affect the speciation, mobility, and bioavailability of arsenic [2]. In water, depending on pH, arsenic and arsenious acid and their deprotonated forms are present. In particular, H_3_AsO_4_ appears up to pH 3, H_2_AsO_4_^−^ is present in the pH range 1–8, HAsO_4_^2−^ in the range 5–13, and, beyond pH 13, only AsO_4_^3−^ exists. Concerning As^III^ species, the neutral H_3_AsO_3_ is the only species that goes up to pH 7, which is when H_2_AsO_3_^−^ starts to generate, and then HAsO_3_^2−^ begins at pH 10 and AsO_3_^3−^ at pH 12 [3]. Generally, natural water is in the range of pH 4–8, while coal and acid drainage are more acidic (pH 2–4) [4].

Over the years, various methods have been set up to remove As^III^ and As^V^, such as precipitation, filtration, reverse osmosis, ion exchange, electrodialysis, and adsorption. Among them, the adsorption process features several advantages, such as high efficiency, low cost, regeneration possibility, and flexibility of operation, even though, only rarely, secondary pollution issues, caused by the sorbent itself, are evaluated [5,6]. The most employed materials include zeolites [7,8], functionalized porous silica [9,10], MOFs [2,11], carbon nanotubes [2,12], and nanoparticles (NPs) [13,14]. The latter ones have found extensive use in the last decades, thanks to the high surface/volume ratio and, therefore, high density of active sites. Iron-based compounds have a strong affinity towards both As^III^ and As^V^. The adsorption properties of iron oxides and oxyhydroxides generally depend on the presence of surface hydroxyl groups. When iron ions on the oxide surface are exposed to bound water, they can complete their coordination shells with hydroxy groups. Depending on pH, these hydroxy groups may bind or release, developing a surface charge that often is associated with adsorption properties governing a particular adsorption mechanism. The arsenic adsorption occurs via ligand exchange of the As species with OH_2_ and OH^−^ in the coordination spheres of the surface structural Fe atoms [15]. Indeed, the arsenate species should link the iron oxide/oxyhydroxide surface through inner-sphere complexation with oxygen bridges, generating a bidentate binuclear complex for akaganeite and ferrihydrite, and both bidentate binuclear and monodentate mononuclear complex for maghemite [16]. Among the various Fe-based systems, the most employed include the oxides maghemite/magnetite (Fe_3_O_4_/γ-Fe_2_O_3_) [17,18,19,20], and hematite (α-Fe_2_O_3_) [21,22], which possess high arsenic adsorbed amount in a wide pH range. Other commonly employed iron compounds are the oxyhydroxides, such as goethite (α-FeOOH) [23], akaganeite (β-FeOOH) [24,25,26,27,28,29,30], schwertmannite [28], and ferrihydrite [23,31,32,33,34], the latter one being the most studied. Thanks to the high number of hydroxyl groups and the generally higher specific surface area, the oxyhydroxides are considered more efficient than the oxides towards both As^V^ and As^III^ [35]. Nevertheless, there is not a direct comparison among iron-based compounds in a wide pH range for the understanding of the different mechanisms involved in arsenic removal.

In this work, the adoption of the same experimental conditions permitted a head-to-head comparison among iron oxides and oxyhydroxides, namely maghemite, akaganeite, and ferrihydrite, for their As^V^ and As^III^ uptake as a function of the pH of the solution. Moreover, the iron oxide was compared with a meso/macroporous silica-iron oxide composite featuring a high surface area. Finally, the akaganeite performance at pH 3 and 8 was deepened through isotherm, kinetics, ionic strength, and competitor studies. Special attention was devoted to the changes in the pH of the solution upon the contact with the sorbents, providing a valuable tool to interpret the adsorption results and the different behavior of the iron oxides and oxyhydroxides towards the arsenicals.

## 2. Materials and Methods

### 2.1. Chemicals

Ethylenediaminetetraacetic acid (98%), poly(ethylene glycol)-block-poly(propylene glycol)-block-poly(ethylene glycol), and sodium (meta) arsenite (90%) were purchased from Aldrich (St. Louise, MO, USA). Iron (III) chloride tetrahydrate (99%) was purchased from Acros Organics (Geel, Belgium). Arsenic (V) standard for ICP (1000 mg L^−1^) was purchased from Fluka (St. Louise, MO, USA). Tetramethyl orthosilicate (98%) was purchased from Merck (Darmstadt, Germany). Sodium arsenate dibasic heptahydrate (98%) was purchased from Sigma (St. Louise, MO, USA). Ammonium hydroxide solution (28–30%) and iron (III) nitrate nonahydrate (98%) were purchased from Sigma-Aldrich (St. Louise, MO, USA). Glacial acetic acid (99.9%), nitric acid (Normatom 67–69%), sodium acetate (99.2%), sodium hydroxide (99.4%), and anhydrous sodium sulphate (99.2%) were purchased from VWR (Leuven, Belgium).

### 2.2. Synthesis of the Sorbents

The akaganeite sample (**Aka**) was synthesized starting from a literature procedure but with some modification [36]. In a 100 mL borosilicate bottle with a polypropylene cap, 12.5 mL of 0.2 M EDTA was added to 28.5 mL of 5.26 M sodium hydroxide solution. To this solution, 25 mL of 2 M FeCl_3_·6H_2_O solution was added (pH = 10) under vigorous stirring. The pH was adjusted to 2 with the addition of HCl 37% *w*/*w*, and the suspension was aged at 98 °C for 4 h in a laboratory oven. The bottle was then rapidly cooled in an ice bath. The solid was separated through centrifugation at 7000 rpm, washed several times with water, and then with ethanol until the chloride content was considered as structural (Cl/Fe = 0.11, estimated by STEM-EDX analysis) [37], then collected and dried under air at 55 °C for two days.

Ferrihydrite (**Fer**) was obtained by adding 180 mL of KOH 5 M to 100 mL 1 M Fe(NO_3_)_3_ solution [38]. The solid was recovered by centrifugation at 7000 rpm, washed several times with water until the removal of potassium ions (K/Fe = 0.003, estimated by STEM-EDX analysis), and dried at 40 °C in the oven for 48 h.

The maghemite sample (**Mag**) was prepared through the oxidation of magnetite in air. Magnetite was synthesized by adapting a co-precipitation method [38], dissolving 4.0590 g of FeCl_2_·4H_2_O in 10 mL HCl 2 M to obtain a solution containing Fe^II^ 2 M. This solution was added in a flask with 50 mL of a 1 M solution of Fe^III^, obtained by dissolving 20.6179 g of Fe(NO_3_)_3_ in HCl 2 M. Then, 500 mL of NH_3_ 1.4 M were added dropwise, using a burette, to the solution of Fe^II^ and Fe^III^ under stirring. The as-formed black precipitate was left to settle for 10 min and separated from the liquid solution using a magnet. The solid was finally washed four times with water and left to dry in an oven at 50 °C.

The silica-iron oxide composite (**Comp**) was prepared from porous silica adapting a method from the literature [39]. Briefly, at 35 °C, 4.6 g of pluronic 123 (P123) and 7.7 g of Na_2_SO_4_ were dissolved in 135 g of 0.02 M acetic acid-sodium acetate buffer solution at pH = 5 for 16 h to form a homogeneous milky mixture under stirring. To this solution mixture, 10.24 mL of TMOS was added under stirring. After 5 min, the stirring was stopped. The resultant mixture was kept in a static condition for 24 h and then transferred into a Teflon-lined autoclave and heated to 100 °C for 24 h. The mixture was centrifuged, and the supernatant discarded. The solid was repeatedly washed with distilled water to remove the inorganic salts and then dried at room temperature. The final product was obtained by calcination under air at 550 °C for 5 h (heating rate 2 °C min^−1^) to remove the organic template. For the impregnation step, 1.0073 g of silica, dried at 120 °C overnight, was dispersed in 25 mL of ethanol and left to homogenize for 1 h under stirring in a crucible. To this mixture, 20 mL of an ethanolic solution of iron nitrate was added under stirring. The mixture was left under a fume hood until most of the ethanol evaporated and a dense paste remained. The crucible was then transferred to a pre-heated furnace at 400 °C for 3 h to decompose the iron nitrate. The final iron oxide content was 28.3% *w*/*w*.

### 2.3. Adsorption Tests

About 50 mg of sample were placed in a 50 mL centrifuge tube with 20 mL of arsenic solution at various concentrations. The solutions were prepared in volumetric flasks, using Milli-Q water, starting from Na_2_HAsO_4_·7H_2_O as the source of As^V^ or NaAsO_2_ as the source of As^III^. The pH of the solutions was modified before contact with the solid sorbent material by adding 0.1 M or 1 M NaOH or HCl. 5 mL of this starting solution were diluted with 5 mL of 4% *w*/*w* HNO_3_ for subsequent ICP-OES analysis, while 20 mL were put in the 50 mL centrifuge tubes containing the solid samples. After contact with the solid, the pH of the mixtures was measured again. The tubes were then put in an orbital shaker, rotating at 40 rpm for 16 h. After centrifugation at 8000 rpm for 10 min, the supernatant was separated and filtered with a 0.45 μm sieve. The pH of the solution was measured again, then 8 mL were transferred into a 15 mL testing tube together with 2 mL of nitric acid 10% *w*/*w* and analyzed by ICP-OES. Several parameters were modified, such as initial pH (pH_0_ 2–8), initial concentration (C_0_ 10–500 mg L^−1^), arsenic oxidation state (As^III^ or As^V^), contact time (10–960 min), ionic strength (NaCl 0.01–1 M), and competitors (SO_4_^2−^ or PO_4_^2−^ 1:100–1:1), as shown in Appendix A.

### 2.4. Isotherm Models

The adsorbed amount of arsenic (*q_e_*) was calculated through Equation (1) after correcting the sorbent mass for the water content, estimated via gravimetric analysis by heating the sample at 105 °C.
(1)$$q_{e}\left(\frac{mg}{g}\right)=\frac{\left(C_{0}-C_{e}\right)V}{m}$$
where *C*_0_ is the initial concentration of As solution (mg L^−1^), *C_e_* is the equilibrium concentration of As in a solution after the batch experiment (mg L^−1^), *V* is the volume of As solution (L), *m* is the amount of adsorbent (*g*). By plotting *C_e_* vs. *q_e_* it was possible to fit the experimental data with the non-linear regression forms of the Langmuir (Equation (2)) [40], Freundlich (Equation (3)) [41], Temkin (Equation (4)) [42], Redlich–Peterson (Equation (5)) [43], and Dubinin–Radushkevich (Equation (6)) [44] isotherm models (Table 1).

In the Langmuir isotherm model (Equation (2)), *q_m_* is the maximum adsorption capacity (mg g^−1^), and *K_L_* is the Langmuir constant (L mg^−1^), which is related to the energy of adsorption. It assumes that each active site is equivalent, and it is energetically irrelevant whether adjacent sorption centers are empty or occupied.
(2)$$q_{e}=\frac{q_{m}K_{L}C_{e}}{1+K_{L}C_{e}}$$

In the Freundlich isotherm model (Equation (3)), *K_F_* is the Freundlich constant, which gives an estimation of the amount of sorbate retained per gram of adsorbent at the equilibrium concentration (mg^1−1/n^ L^1/n^ g^−1^), and *n* is a measure of the nature and strength of the sorption process and the distribution of active sites related to the surface heterogeneity (the heterogeneity of the system increases with *n*). Therefore, it assumes that the sorption process occurs on non-equivalent active sites due to repulsion between sorbent species.
(3)$$q_{e}=K_{F}C_{e}^{1/n}$$

In the Temkin isotherm model (Equation (4)), *b_T_* (J g mol^−1^ mg^−1^) and *K_T_* (L mg^−1^) are parameters describing the adsorbate-adsorbent interactions.
(4)$$q_{e}=\frac{RT}{b_{T}}ln(K_{T}C_{e})$$

It assumes that the heat of adsorption decreases linearly with the increase in the amount of adsorbed species.

The Redlich–Peterson isotherm model (Equation (5)) is a hybrid between the Langmuir and Freundlich models.
(5)$$q_{e}=\frac{K_{RP}C_{e}}{1+\alpha_{RP}C_{e}^{\beta_{RP}}}$$
where *K_RP_* (L g^−1^), *α_RP_* (L mg^−1^), and *β_RP_* are the Redlich–Peterson parameters.

In the Dubinin–Radushkevich isotherm model (Equation (6)), *ε_DR_* (kJ mol^−1^), and *K_DR_* (mol^2^ kJ^−2^) are the Dubinin–Radushkevich isotherm variable and constant, respectively.
(6)$$q_{e}=q_{m}e^{\left(-K_{DR}\varepsilon_{DR}^{2}\right)}$$

The model is used to differentiate between physisorption and chemisorption. The mean free energy of adsorption *E_ads_* (kJ mol^−1^) can be calculated following Equation (7).
(7)$$E_{ads}=\frac{1}{\sqrt{2K_{DR}}}$$

### 2.5. Kinetic Models

The adsorbed amount of arsenic at a certain time (*q_t_*) was calculated through Equation (8).
(8)$$q_{t}=\frac{\left(C_{0}-C_{t}\right)V}{m}$$

The plotted data *q_t_* vs. *C_t_* were then fitted by the pseudo-first order (Equation (9)) and pseudo-second order (Equation (10)) kinetic models (Table 2).
(9)$$q_{t}=q_{e1}\left(1-e^{\left(K^{\prime }t\right)}\right)$$
(10)$$q_{t}=\frac{K^{\prime \prime }q_{e2}^{2}t}{1+K^{\prime \prime }q_{e2}t}$$
where *K*′ (min^−1^) and *K*″ (g mg^−1^ min^−1^) are the pseudo-1st order and pseudo-2nd order constants, respectively. The pseudo 2nd order model in linearized form (Equation (11)) was then used to fit the *t/q_t_* vs. *t* plots.
(11)$$\frac{t}{q_{t}}=\frac{1}{K^{\prime \prime }q_{e2}^{2}}+\frac{t}{q_{e2}}$$

The kinetic data were fit also by the intraparticle diffusion model (Equation (12)).
(12)$$q_{t}=k_{i}t^{\frac{1}{2}}+x_{i}$$
where *k_i_* is the intraparticle diffusion constant (mg g^−1^ min^−1/2^)

### 2.6. Characterization Techniques

The solutions were analyzed by Inductively Coupled Plasma-Optical Emission Spectrometry (ICP-OES) using an Agilent 5110 device (Agilent, Santa Clara, CA, USA). The calibration line was performed in the range 1–100 mg L^−1^ at wavelength 188.980 nm for arsenic. Each sample was analyzed three times in 2% *w*/*w* HNO_3_ solution. The samples **Fer**, **Mag**, and **Comp** were characterized by powder X-ray Diffraction (XRD) using a PANalytical X’pert Pro (Malvern PANalytical, Malvern, UK) equipped with Cu Kα radiation (1.5418 Å). The sample **Aka** was analyzed through a Rigaku Smartlab diffractometer (Rigaku Corporation, Tokyo, Japan) equipped with a 9 kW rotating anode and graphite monochromator in the diffracted beam with Bragg–Brentano parafocusing geometry. The refinement of the structural parameters was performed by the Rietveld method using the MAUD software (v 2.991, Radiographema, Trento, Italy) [45] and LaB_6_ from NIST as a reference standard for determining the instrumental parameters. The CIF structure used for the refinement were 0003079 from AMCSD for akageneite [46], 9011571 from COD for ferrihydrite [47], and 9006316 from COD for maghemite [48]. Room Temperature (RT) ^57^Fe Mössbauer spectroscopy was done on a Wissel spectrometer (Wissenschaftliche Elektronik GmbH, Stamberg, Germany) using transmission arrangement and proportional detector LND-45431. An α-Fe foil was used as a standard, and the fitting procedure was done by the NORMOS program (v 25.1.1989, University of Duisburg-Essen, Duisburg, Germany). Transmission electron microscopy (TEM) images were obtained using a JEOL JEM 1400 Plus (Jeol Ltd., Tokyo, Japan) operating at 120 kV. The specimens were prepared by dropping an ethanol dispersion of the samples on a 200-mesh carbon-coated copper grid. High-Resolution TEM images were carried out through a JEOL JEM 2010 UHR (Jeol Ltd., Tokyo, Japan) operating at 200 kV equipped with a 794 slow-scan CCD camera. Zeta potential measurements were performed through a Malvern Instrument Zetasizer Nano ZSP (Malvern PANalytical, Malvern, UK) equipped with a He-Ne laser (λ = 633 nm, max. 5 mW) and operated at a scattering angle of 173°, using Zetasizer software (v 7.03, Malvern PANalytical, Malver, UK) to analyze the data. The sample was prepared by suspending the composites (5 mg mL^−1^) in distilled water and adding HCl and NaOH to modify the pH from 2 to 9. The scattering cell temperature was fixed at 25 °C. Textural analyses of all samples were performed on a Micromeritics ASAP 2020 (Micromeritics, Norcross, Georgia, USA) by determining the nitrogen adsorption−desorption isotherms at −196 °C. Prior to analyses, the iron oxides and hydroxides samples were heated for 12 h under a vacuum at 120 °C (heating rate, 1 °C min^−1^), while treatment at 250 °C (heating rate, 1 °C min^−1^) for 12 h was applied on the bare silica and silica-composite sample. The specific surface area (S_BET_) was computed by the Brunauer−Emmett−Teller (BET) equation [49] from the adsorption data in the P/P_0_ range 0.05−0.30 for the mesoporous samples **Aka**, **Mag**, **Silica**, and **Comp**, while the Dubinin–Radushkevic model [44] was applied in the sample **Fer**, due to its own microporous nature. The total pore volume (V_p_) was calculated at P/P_0_ = 0.87. The pore diameter was determined by applying the Barrett−Joyner−Halenda (BJH) model [50] to the isotherm desorption branch for the mesoporous samples **Aka**, **Mag**, and **Comp**, while the Horvath-Kawazoe model [51] was adopted for the microporous **Fer**. FTIR spectra of the sorbents were acquired in a KBr pellet through a Bruker Equinox 55 spectrophotometer (Bruker, Billerica, MA, USA) in the region 400–4000 cm^−1^. The spectra were processed with OPUS software (v 7.6, Bruker, Billerica, MA, USA). The sorbents, after arsenic uptake, were analyzed by means of an Agilent Cary 630 spectrophotometer (Agilent, Santa Clara, CA, USA) equipped with an ATR module in the range 650–4000 cm^−1^. The spectra were processed with Microlab PC (v 5.5.1989, Agilent, Santa Clara, CA, USA).

## 3. Results and Discussion

### 3.1. Characterization of the Sorbents

The Fe^III^-based sorbents were prepared via easy and low-cost methods to obtain nanosized systems. 

XRD and RT ^57^Fe Mössbauer spectroscopy (Figure 1a,c) show that all the iron oxide samples feature a single Fe^III^-based structure. Monoclinic I2/m akageneite is ascribed to **Aka** and cubic Fd3m maghemite for **Mag**. **Fer** sample displays the typical pattern of two-lines ferrihydrite, and it was fitted with the hexagonal P63mc phase. **Comp**, on the contrary, reveals a broadband at about 22°, typical of amorphous silica, and the distinctive reflexes of two Fe^III^ oxides, i.e., hematite and maghemite. All the RT Mössbauer spectra (Figure 1c) are characterized by isomer shift values in the range 0.32–0.38 mm s^−1^, typical of Fe^III^-based phases (Appendix A) [52,53,54,55,56,57,58]. The **Aka**, **Fer**, and **Comp** spectra feature one or more doublets, whereas the **Mag** spectrum can be fitted with two broad sextets, accounting for the distribution of hyperfine fields. The two sextets feature hyperfine field values of 47.09 (3) and 41.9 (4) T, corresponding to iron cations in the tetrahedral and octahedral sites of the spinel ferrite structure, respectively. The isomer shift for both the sextets is in the range 0.32–0.34 mm s^−1^, typical for Fe^III^, indicating the effective oxidation of Fe^II^ of magnetite from which it derived, whose values are around 0.6–0.7 mm s^−1^. In the case of **Aka**, the spectrum was fitted with two doublets, as suggested in the literature [38], leading to isomer shift values of about 0.37 mm s^−1^ and quadrupole splitting of 0.536 (7) mm s^−1^ and 0.940 (9) mm s^−1^, respectively. The spectrum of **Fer** can be fitted, based on a previous study [59], with three doublets corresponding to different non-equivalent iron positions in the ferrihydrite structure, as reported in Appendix A. The **Comp** spectrum was fitted with a doublet with isomer shift equal to 0.34 (1) mm s^−1^ and quadrupole splitting of 0.74 (2) mm s^−1^, similar to those obtained for similar systems of maghemite/hematite NPs impregnated in porous silica matrixes [60,61].

The small-angle X-ray patterns of the silica-based samples (Figure 1b) show the presence of a shoulder at about 1.5°, which indicates the presence of an ordered porous structure in the mesoporous range [60,61].

FTIR spectra of the samples (Figure 1d) reveal the typical bands of iron oxides and oxyhydroxides (Appendix A), besides those related to water. In particular, **Aka** shows two Fe-O vibrational modes at 680 and 470 cm^−1^ [24,29,62]. The band at 570 cm^−1^ and shoulders at 820 and 630 cm^−1^ in the sample **Mag** are a clear indication of the presence of maghemite, in agreement with ^57^Fe Mössbauer data [38,55,63]. For the sample **Fer**, the Fe-O band is placed at about 600 cm^−1^, while the bands at 1500 and 1330 cm^−1^ are related to the Fe-OH stretching modes [38]. The sample **Comp** discloses the bands associated with silica (Si-O-Si stretching modes at 1220, 1090, and 465 cm^−1^, and Si-OH stretching mode at 810 cm^−1^), whereas those related to the iron oxide phase are difficult to be detected probably because of its form as nanocomposite [60,61]. 

The textural properties of the sorbents were studied through N_2_ physisorption (Figure 1e, Appendix A, Table 3). **Aka**, **Mag**, and **Comp** present an IV-type isotherm with an H1 hysteresis loop characteristic for mesoporous materials. On the contrary, **Fer** features an I-type isotherm with a H3 hysteresis loop typical of microporous materials. As expected, the largest surface area (410 m^2^ g^−1^) is observed for **Comp**, followed by **Fer**, **Aka**, and **Mag** (from 92 to 260 m^2^ g^−1^). The pore volumes are instead higher for the mesoporous materials (**Comp** > **Aka** > **Mag**) and lower for **Fer** due to the presence of only micropores. The pore size distributions (PSD) of **Mag** and **Aka** (Figure 1f) are centered at about 11.8 and 9.4 nm, respectively, while sharper PSD is observed for **Comp** due to the mesostructured nature of the silica matrix. For **Fer**, the micropore distribution, obtained using the Horvath–Kawazoe model, showed a maximum at about 0.7 nm. The comparison between **Comp** and the bare silica matrix (Appendix A), reveals a decrease of 10% of surface area and 11% of pore volume in the first one, as expected after the impregnation process. The PSD is instead centered, for both samples, at about 8 nm, suggesting the formation of isolated NPs inside the pores in spite of a uniform layer [39,60,61].

The differences in the surface areas and the pore volumes observed among the iron oxide/oxyhydroxides are mainly due to the morphological properties of the samples in terms of the size and shape of NPs. For this reason, TEM analyses were conducted on all samples and are shown in Figure 2. In the case of **Aka** (Figure 2 and Appendix A), nanorods of about 60 nm in length and 4 nm in width are observed. **Fer** (Figure 2c) reveals aggregates of small NPs of about 4 nm, while **Mag** (Figure 2d) is composed of spheroidal NPs of about 12 nm. The silica-based samples are constituted of both ordered mesopores in the range 7–9 nm (white arrows Figure 2f) and macropores of about 150 nm (Figure 2e,f), present also after the impregnation step (Figure 2g,h). Moreover, some dark spots of about 10 nm, corresponding to the iron oxide NPs, are visible inside the mesopores of **Comp**, with no evidence of particles outside the matrix (white arrows in white arrows in Figure 2h).

The HRTEM micrographs of the **Aka** sample (Figure 2b and Appendix A) confirm the crystallinity of the particles and reveal the crystalline planes typical of akaganeite, i.e., {301}. In some cases, it is possible to observe the formation of nanotubes (Figure 2a inset, white arrows Appendix A). Moreover, as can be seen in Figure 2a and Appendix A, some small NPs of about 3 (1) nm are visible.

The Rietveld refinements of the XRD patterns (Appendix A) were performed on the basis of the information extracted from TEM analysis. The cell parameters, the crystallite sizes, and the relative fraction of the phases (for **Comp** and **Aka**) were determined (Table 3). The XRD pattern of **Aka** was refined by using two populations of akageneite particles: one referring to isotropic particles and one to those with anisotropic shape, for which isotropic and anisotropic-no-rules models [64] were used, respectively. A diameter for the isotropic model of 5.3 (1) nm was found, while, for the anisotropic one, a minimum dimension (D_XRD1_) of 2.0 (6) nm and a maximum one (D_XRD2_) of 25.1 (2) nm were obtained, corresponding to the D_11_ and D_22_ textural components, respectively. The lower crystallite size values, in comparison with those obtained for the particles by TEM, are probably derived from the presence of NPs made up of at least two crystallites close to each other. For **Mag** and **Fer**, an isotropic model was employed since it gave satisfactory outcomes, resulting in crystallite sizes of 14.0 (1) nm and 1.7 (3) nm, respectively, in good agreement with the TEM observations. **Comp** was found to be composed of 18% *w*/*w* of hematite and 82% *w*/*w* of maghemite, both featuring crystallite sizes between 7 and 9 nm, compatible with the mesopore size of the matrix.

In view of possible applications as adsorbents for ionic species from polluted water, the evaluation of the surface charge of the samples at different pH is crucial. Therefore, the zeta (ζ) potential measurements on all samples (Figure 3) were performed. **Comp** features the lowest surface charge at acidic pH (≈5 mV) and the lowest isoelectric point (pI ≈ 4.5), mainly due to the high amount of silica, which features a low surface charge [65]. On the contrary, **Mag** presents higher ζ-potential values (>20 mV) up to pH 5 and pI ≈ 7. A higher isoelectric point is observed for **Fer** (pI ≈ 8.5) and **Aka** (pI ≈ 10), together with higher ζ-potential values when the surface is positively charged up to pH 7 (30–40 mV). 

### 3.2. Effect of the pH in the As^V^ and As^III^ Test Removal by Fe^III^-Based Sorbents

To estimate the optimal pH value for the adsorption, the first experiment focused on the pH dependence of the As^V^ and As^III^ adsorption capacity of the adsorbents. Indeed, this process depends on the arsenic species present in the solution, as can be seen in the Bjerrum plot in Appendix A, and on the surface species and charge of the different sorbents as a function of the pH (Figure 3 and Appendix A), therefore several reactions are possible (Appendix A).

Different initial pH conditions were tested, namely pH_0_ 2, 3, 4, 6, and 8, at 100 mg L^−1^, employing 50 mg of sorbent and 20 mL of arsenic solution (Figure 4a, Appendix A).

For As^V^, **Aka** is the most efficient one, with a removal capacity close to 100% in the whole pH_0_ range. At pH_0_ 2 and 3, **Fer** also features high arsenic uptake (100% and 94%, respectively), but its efficiency drops to 56% at pH_0_ 4 and 50% at pH_0_ 6, finally reaching 23% at pH_0_ 8. A similar behavior, but with a more gradual worsening and lower performance, is observed for **Mag** (pH_0_ 2: As^V^ removal = 68.5%; pH_0_ 8: As^V^ removal = 16.2%) and **Comp** (pH_0_ 2: As^V^ removal = 26.1%; pH_0_ 8: As^V^ removal = 6.1%). The pH measure of the arsenic solution before contact with the sorbents (pH_0_), immediately after the contact (pH_Int_), and after the batch tests (pH_Fin_) reveals interesting information about the adsorption process (Appendix A). For **Aka**, a decrease in pH is observed, more consistent as pH_0_ increases, while for **Fer** and **Mag**, an opposite trend can be depicted, with an increase of pH immediately after the contact of the solid with the solution, in particular at pH_0_ 4. For **Comp**, similar behavior is observed with the exception of pH_0_ 6 and 8, at which a decrease in the pH is observed. To discern whether the arsenic species or the sorbents themselves were the cause of the pH modification, all the sorbents were put in contact with water (pH ≈ 5.5), and the pH was measured immediately after. As can be seen in Appendix A, the pH of the solution containing **Aka** decreased to 3.09, probably due to the diffusion of Cl^−^ and H^+^ from the akaganeite channels toward the solution [30,66]. Other authors reported an opposite trend with an increase in the arsenic solution pH from 3.5 to 6, due to the contact with the akageneite, but, unfortunately, no explanation was provided [24]. On the contrary, the other sorbents did not cause drastic pH changes. Indeed, only a slight increase was observed for **Fer** (pH 6.08) due to protonation of the surface hydroxyl groups by water molecules to form ≡FeOH_2_^+^ and consequent release of OH^−^ (Appendix A) [38]. On the contrary, **Mag** and **Comp** displayed a pH decrease (4.95 and 5.21, respectively), caused by the Lewis acid behavior of unsaturated surface Fe atoms in the first case [38], and the formation of ≡SiO^−^ in the latter one, both accompanied by a release of H_3_O^+^ (Appendix A) [67].

Considering the different behaviors of the sorbents in modifying the pH, the iron oxide and oxyhydroxides were put in contact with different As-free solutions to study the evolution of the pH (pH_Int_) and the zeta potential (Appendix A). The solutions were prepared by adding HCl or NaOH to Milli-Q water to obtain the pH_0_ 2, 3, 4, 6, and 8. Also in this case, **Aka** caused a pH reduction in the whole range, accompanied by a high and positive zeta potential (36–43 mV). **Mag** did not induce any substantial pH modification up to pH_0_ 4, but at pH_0_ 6 and 8, a pH reduction to 5.30 and 6.11, respectively, was observed, with zeta potential values in the range 30–21 mV. Concerning **Fer**, at pH_0_ 3, 4, and 6, an increase of pH to 5.58, 7.06, and 7.33, respectively, was depicted. In this case, contrary to what was observed in Figure 3, the zeta potential fell down to 0 mV, already at pH_Int_ 7.06 and assumed negative values at pH_Int_ 7.33 (−33 mV). This discrepancy can be ascribed to the lower ionic strength in this latter experiment that does not permit the formation of an electric double layer [38]. Indeed, in the tests reported in Figure 3, the sorbent dispersion pH was modified firstly with HCl down to pH 2, then increased with NaOH. The higher amount of Na^+^ adsorbed on the sorbent surface is permitted to have higher zeta potential values, confirming the role of the adsorbed ions in the sorbent properties and behavior.

The above discussion permits us to understand better the role of sorbents in arsenate removal. Indeed, the pH reduction for **Aka**, observed during the As^V^ uptake tests, is caused not by the removal of arsenate but by the sorbent itself. In the case of **Fer**, **Mag**, and **Comp,** the change in the pH during the As^V^ batch tests, is caused both by the surface chemistry of the sorbent and the arsenate solution equilibria. In fact, the increase in the pH, starting from pH_0_ 3, is due to the protonation of the surface hydroxyl groups, but also involves the removal of arsenic species, as evidenced by the differences between pH_Int_ and pH_Fin_. On the contrary, the decrease in the pH for **Comp** at pH_0_ 6 and 8 can be mainly ascribed to the deprotonation of the silica surface, since only a low amount of arsenate species is removed. 

Therefore, it is worth noting that the pH, in order to estimate the surface charge and the arsenic speciation, is derived from the contact of the arsenic solution with the sorbent (pH_int_ in Appendix A), which in many cases differs from the initial pH value (pH_0_ in Appendix A). In this optic, the decrease in the As^V^ uptake with the increase in pH_int_ agrees with the observed trends of the ζ-potential: a positive charge is found at acidic pH based on dominant ≡FeOH_2_^+^ species, and a negative charge is found at basic pH due to a majority of superficial ≡FeO^−^. This determines a different extent of interaction between the sorbent surface and the arsenate anions as a function of the pH [38]. The higher efficiency of **Aka**, featuring 100% of As^V^ removal in the whole pH_0_ range, can be explained considering the pH_Int_ instead of pH_0_ since the sorbent itself drops it down to more acidic pH, where the oxyhydroxide is positively charged and works better (Appendix A). For **Fer**, only at pH_0_ 2 and 3, pH_Int_ remains acid, while for the other pH_0_ values, neutrality or basicity was observed after the sorbent–As^V^ solution contact. The decrease in As^V^ uptake at pH_0_ 8 can be easily explained considering the negative charge of the **Fer** surface (Figure 3). At pH_0_ 4 and 6, corresponding to pH_Int_ 6.5–6.8, we must consider that besides H_2_AsO_4_^−^, HAsO_4_^2−^ is also already present in the solution (Appendix A), whose adsorption on the oxyhydroxide surface is less favored due to the release of worse leaving groups than those for H_2_AsO_4_^−^ (Appendix A, reactions +2A/+2B vs. +3A/+3B). A comparison between **Fer** and **Mag** reveals that pH_Fin_ is always higher for the first sorbent, with a different trend with respect to pH_Int_. 

Since the two oxyhydroxides featured the best performances at 100 mg L^−1^ toward As^V^ removal, the tests were repeated employing 500 mg L^−1^ as the starting concentration (Appendix A, Figure 4b, Appendix A), to evaluate the pH-dependence in sorbent saturation condition. It is worth noting that, for **Aka**, pH_int_ was found to be quite close to pH_0_, due to the high concentration of arsenate species, which act as a buffer solution. In this case, as for the other sorbents, it is possible to observe a decrease in As^V^ uptake with increasing the pH_0_, with adsorption capacity equal to 87 mg g^−1^ at pH_0_ 2 and 51 mg g^−1^ at pH_0_ 8. However, this decrease is not gradual, with almost constant values observed between pH_0_ 2 and 4 and a higher worsening of the performances at pH_0_ 6 and 8. If the surface charge is considered, one should expect a constant behavior of up to pH_0_ 7, while we observed a drop already at pH_0_ 6, as explained above due to the presence of HAsO_4_^2−^ [15]. Moreover, at basic pH, the OH^−^ present in solution competes with the negatively charged arsenate species (Appendix A) [25], lowering the As uptake. For **Fer**, the trend of pH_Int_, pH_Fin_, and As^V^ removal, is similar to what was observed at 100 mg L^−1^. The results revealed a maximum adsorption capacity reached at pH_0_ 2 equal to 71 mg g^−1^, lower than that of **Aka** (87 mg g^−1^). If *q_e_* values are normalized for the surface areas (Table 3), the arsenic uptake of **Aka** and **Fer** are 0.43 and 0.27 mg m^−2^, respectively, probably caused by the preferential orientation of akaganeite nanotubes along specific directions, which can have higher concentration of active sites. In addition, this study afforded the same experimental conditions and confirmed the higher efficiency of both oxyhydroxides (Figure 4), if compared to oxides, due to the higher density of superficial hydroxyl groups and surface area [35]. Finally, if **Mag** and **Comp** are compared by normalizing the *q_e_* values for the active phases (Appendix A), their efficiency is similar, and in some cases higher for **Comp**, indicating the complete accessibility of the iron oxide inside the pores. Indeed, the ideal advantage in dispersing an active phase in porous silica may reflect higher chemical and mechanical stability and the possibility to modify the silica walls with other kinds of functional groups and/or active inorganic phases [35,68]. Conversely, one should evaluate the cost of producing such sorbents and the possible issues related to secondary silicon pollution [39]. As reported in Appendix A, silicon was found after the adsorption tests, with concentrations that increase with the pH.

Regarding the adsorption of As^III^ (C_0_ = 100 mg L^−1^), all the samples display a lower arsenic removal at pH_0_ 2, then an increase and a steady behavior in the pH_0_ range 3–8 (Figure 4b, Appendix A), as already observed in the literature for akageneite in this pH range [29]. The different behavior, if compared to As^V^, is explained by the existence of neutral species (H_3_AsO_3_) up to pH 8, whose uptake is not affected by the surface charge of the sorbents (Appendix A) [25]. In this range, the most efficient sample becomes **Fer**, having removals close to 96% and an adsorbed amount of about 50 mg g^−1^, higher than that of Aka (As^III^ removal 80%, *q_e_* = 36 mg g^−1^). This result indicates that arsenious acid does not diffuse well inside **Aka** nanotubes, probably due to the absence of attractive electrostatic forces, indicating that not all of the akaganeite surface is available for As^III^ uptake, in contrast with **Fer**. The evaluation of the effect of the contact between the sorbents and the As^III^ solution on the pH (Appendix A) revealed a similar behavior when compared to the As^V^ one, with some differences. For instance, the pH decrease, for **Aka**, at pH_0_ 8 is more significant, probably due to the absence of buffer effects from arsenite species (Appendix A). For, **Fer**, **Mag**, and **Comp**, the discrepancy between the pH values is less important. Only small differences can be identified in the comparison with the As^V^ adsorption. For instance, at pH_0_ 8, a decrease in pH_Int_ is visible, caused by the iron oxide itself (Appendix A). 

The adsorbed amount normalized for the active phase for the sample **Comp** is lower if compared to **Mag**, contrary to what was observed for As^V^ removal (Appendix A). Again, this result can be justified considering that the diffusion of As^III^ species through the silica mesochannels is not favored due to the absence of attractive electrostatic forces, similar to what was observed for **Aka**. Concerning the secondary silicon pollution (Appendix A), the comparison between As^III^ and As^V^ tests reveals that the phenomenon is limited in the case of the As^III^ species, and the silicon release is mainly affected by the pH, probably due to a weaker interaction of arsenite with the sorbent. On the contrary, the Si release observed for the As^V^ removal tests indicated that arsenate species play a crucial role, as already observed in a previous study [39], beyond a pH effect.

As for As^V^ uptake, the As^III^ removal was studied under sorbent saturation condition (C_0_ = 500 mg L^−1^) for the two oxyhydroxides (Figure 4d, Appendix A). Similar results to the 100 mg L^−1^ tests were found in the arsenic uptake (but with a higher absolute adsorbed dose), in the pH_Int_ and pH_Fin_ trends, as a function of the pH_0_. The maximum values were reached at pH_0_ 8, equal to 91 mg g^−1^ and 144 mg g^−1^ for **Aka** and **Fer**, respectively. The adsorbed amount normalized for the surface area results higher for **Fer** with respect to **Aka** in the whole pH range, supporting the idea of a reduction in the available surface-active sites for **Aka** due to the absence of diffusion of arsenious acid inside the nanotubes. 

FTIR spectra acquired after As^III^ and As^V^ adsorption on **Aka** and **Fer** are reported in Appendix A. For **Aka**, the libration OH-Cl at 845 cm^−1^ becomes less visible due to the appearance of a new band associated with the As-O stretching at 815 cm^−1^ [25,26,34]. Concerning **Fer**, the As-O band is located at 790 cm^−1^, indicating weaker binding if compared with **Aka**. Furthermore, there is a strong reduction of the bands at 1500, 1330, 1065, and 850 cm^−1^, ascribed to Fe-OH (Appendix A), upon As adsorption. For both **Fer** and **Aka**, the As-O stretching band for As^V^ adsorption is stronger than As^III^, probably due to the involvement of a different number of As-O bonds [16]. 

Hence, even though in the literature there are studies devoted to As^V^ and/or As^III^ removal by both ferrihydrite [23,31,32,33,34] and akageneite [24,25,26,27,28,29], the differences in the experimental conditions hinder a comparison between them. Therefore, the evaluation of the most efficient oxyhydroxide is not straightforward, and, to the best of our knowledge, the current work is the first example of a direct comparison. Even though **Aka** features an As^III^ uptake lower than that of **Fer**, it can be considered the most promising sample. In fact, it should be noted that the amount of As^III^ is generally much lower than that of As^V^ in aerobic environments [68]. Moreover, **Aka** can efficiently remove both As^III^ and As^V^ species in the whole pH_0_ range (2–8).

### 3.3. Effect of Initial Concentration and Isotherm Modelling on the Adsorption of As^V^ by Akaganeite

To deepen the arsenic removal mechanism for the most promising sample, **Aka**, the adsorption of the As^V^ species, which is more sensitive to pH with respect to As^III^ ones in the pH range 3–8 (Appendix A), was studied under different initial concentrations (10–500 mg L^−1^), contact time (10–960 min), ionic strength (NaCl 0–1M), and presence of competitors (sulphate, phosphate) at pH_0_ 3 and 8 (Figure 5, Figure 6 and Figure 7). Concerning the initial As^V^ concentration effect, both at pH_0_ 3 and 8, it is possible to observe a sharp increase in the adsorbed dose and then an almost steady behavior (Figure 5). When pH_0_ is 8, the pH_int_ drastically decreases to 3 for C_0_ = 10 and 50 mg L^−1^ (Appendix A). As the initial concentration increases, the pH drop is less critical due to the buffer effect of the arsenate species present at higher concentrations. For instance, at 150 mg L^−1^ the pH goes down from 8 to 6.5, and at 250 mg L^−1^ from 8 to 7.3. Consequently, the adsorbed dose is lower if compared to the tests made at pH_0_ = 3. The adsorbed dose vs. the equilibrium concentration (*q_e_* vs. C_e_) plot was fitted with different isotherm models, namely Langmuir, Freundlich, Temkin, Redlich–Peterson, and Dubinin–Radushkevich, as described in the experimental section. The parameters are reported in Table 4.

If R^2^ values are considered, the *q_e_* vs. *C_e_* tendency is better described, for both pH values, by the Redlich–Peterson model, which is a hybrid between the Langmuir and Freundlich models, accounting for energetically equivalent or non-equivalent binding sites on the sorbent active-phase, respectively. In the literature, some articles reported isotherms fitted by the Langmuir model in an equilibrium concentration range of 0–70 mg L^−1^ at a pH range close to neutrality [24,26,30,35]. Nevertheless, some of these authors underlined that both the models are appropriate to describe the As^V^ adsorption on akageneite, with only slight differences in the obtained R^2^ values [24]. Moreover, other works report the Freundlich model to best describe the isotherm adsorption of As^III^ on akageneite [29], or As^III^/As^V^ on ferrihydrite [29,31,32]. Therefore, our results (i.e., better fit by Redlich-Peterson model) suggest that both monolayer adsorption and heterogenous surfaces may coexist. A possible alternative interpretation for the isotherm at pH_0_ 3 consists in a change of the sorbent surface during the adsorption process as a function of the concentration. Indeed, up to a certain critical concentration (i.e., C_0_ = 250 mg L^−1^, *C_e_* = 75 mg L^−1^) the best-fitting isotherm model is the Langmuir one, indicating the filling of free, energetically equivalent active sites. Then, for higher concentrations, a better agreement of the experimental data with the Freundlich/Temkin ones is observed, coherent with the formation of adsorbate multilayers or non-energetically equivalent As-O-Fe bonds. This phenomenon is not visible when the pH of the starting solution is 8, where the experimental data well follows the Langmuir model, and probably it is missed in the literature due to the differences in the investigated equilibrium concentration range and pH. Despite the Redlich–Peterson model seems to be the most suitable, the maximum loading estimated by the Langmuir (and Dubinin–Radushkevich) model is about 80 mg g^−1^ at pH_0_ 3 and 50 mg g^−1^ at pH_0_ 8 (Table 4), indicating a higher efficiency at acidic pH, in agreement with the results previously presented. Nevertheless, a higher removal was achieved at pH_0_ 3 for the highest initial As^V^ concentration (89 mg g^−1^, Appendix A) that again can be justified by the presence of non-equivalent active sites, not described by the Langmuir model. 

The FTIR spectra of **Aka** (Appendix A) reveal how the As-O stretching band becomes more intense as the starting arsenic concentration increases from 100 mg L^−1^ to 250 mg L^−1^, while the Fe-OH stretching band at 1360 cm^−1^ disappears [26,34]. 

### 3.4. Effect of Contact Time and Kinetic Modelling on the Adsorption of As^V^ by Akaganeite

The adsorption kinetics were studied at pH_0_ 3 and 8, in the contact time range 10–960 min, employing a starting As^V^ concentration of 250 mg L^−1^ (Appendix A). This As^V^ concentration was chosen to be high enough for the arsenate buffer to resist the pH drop caused by the sorbent at pH_0_ 8, but not too much to generate multilayer phenomena (monolayer sorbent saturation condition). The adsorbed dose at a specific time *versus* time plots (*q_t_* vs. *t*, Figure 6a) were fitted with the pseudo 1st and 2nd models, the latter one better fitting the experimental data, as also evidenced by the linearized plots in Figure 6b and Appendix A. The equilibrium adsorbed amount (q_e_^calc^), obtained from the fitting with the pseudo 2nd model, is close to the experimental one, obtained already after 120 min of contact time (Appendix A). Moreover, after 10 min, 80% of the removable arsenic is already adsorbed for the pH_0_, indicating rapid reactions (Appendix A). The *q_t_* vs. t^1/2^ plots (Figure 6c) were fitted by the intraparticle diffusion model in two different steps, which account for two different adsorption mechanisms. The first one is associated with a faster adsorption process (diffusion of arsenate from the solution to the **Aka** surface), featuring the highest constant at both pH_0_ (*k_i_*) and ending at about 60–120 min. The second step, almost parallel to the x-axis, corresponds to a slower uptake that takes place once the sorbent surface is enriched by arsenate species.

### 3.5. Effect of Added Salts as Competitors the Adsorption of As^V^ by Akaganeite

Since it is known that ionic strength affects the adsorption capacity, tests at pH_0_ 3 and 8, in monolayer sorbent saturation condition (C_0_ = 250 mg L^−1^), were performed by varying NaCl concentration in the range 0–1 mol L^−1^ (Appendix A, Figure 7a). For pH_0_ 3, the As^V^ uptake was almost constant, with just a small decrease with the increase in the NaCl concentration (−7 mg g^−1^). On the contrary, a slight increase was observed at pH_0_ 8 that increased the ionic strength (+14 mg g^−1^). This behavior was also observed by other authors [24], who hypothesized an increase in the surface charge due to the adsorption of cations (K^+^ in their case instead of Na^+^) at basic pH, and a consequent increase in arsenate removal capacity. This phenomenon does not occur at acidic pH due to the repulsion between the superficial ≡FeOH_2_^+^ species and the cations in the solution. On the contrary, the attraction of anions from the solution might occur, leading to a slight worsening of the removal performance. The pH was also slightly affected by NaCl in the solution (ΔpH = +0.1 for a change of one order of magnitude in the molarity), regardless of the presence of arsenate species (Appendix A). This change is strictly related to the chloride ions since the presence of NaNO_3_ did not affect the pH in the same way (Appendix A). Indeed, the presence of chloride in the solution hinders the release of Cl^−^ and H^+^ from the akaganeite channels toward the solution [30,66]. 

With the aim of monitoring the As^V^ uptake with the presence of competitors, sulphate and phosphate were tested at different concentrations, in 1:1, 10:1, and 100:1 molar ratios with respect to arsenate, corresponding to 0.003, 0.033, and 0.334 mol L^−1^ of competitor concentration, respectively (Appendix A). The tests were conducted at both pH_0_ 3 and 8, with an initial arsenic concentration equal to 250 mg L^−1^. The results (Figure 7b,c) show that sulphate features at pH_0_ 8 have a similar behavior to what was observed for NaCl, with a slight improvement (+8 mg g^−1^), but a higher adsorption decrease at pH_0_ 3 is observed (−27 mg g^−1^), probably due to the doubled charge of sulphate anions with respect to chlorides. Conversely, the phosphate causes a drastic decrease in arsenic removal capacity at both pHs (−70 mg g^−1^ at pH_0_ 3, −48 mg g^−1^ at pH_0_ 8), as already observed [30]. This reduction is due to the chemical similarities between phosphate and arsenate for the superficial akaganeite active sites that should create a strong bond through inner-sphere complexation. On the contrary, outer-sphere complexes featuring water molecules between ligands and metal ions are found in the case of sulphate and chloride, which do not strongly influence arsenic adsorption. The presence of competitors also influenced the pH after the adsorption test (pH_fin_). In the case of sulphate, there is no substantial change whether this ion is present or not, and a decrease in pH is observed. When phosphate is employed, its buffer effect stabilizes the pH, avoiding the decrease [27,30]. 

FTIR spectra of the sorbents after the tests reveal the presence of the bands associated with As-O (813 cm^−1^), P-O (1030 cm^−1^), and S-O (1112 cm^−1^), and the disappearance of the Fe-OH band at 1360 cm^−1^ (Appendix A). 

## 4. Conclusions

In this work, a head-to-head comparison of the As^V^ and As^III^ removal ability of iron oxyhydroxides (akaganeite and ferrihydrite) and oxides (Fe_2_O_3_ in the form of NPs and dispersed in a meso/macroporous silica matrix) in the pH range 2–8 are provided. Emphasis was devoted to studying the arsenic solution pH before the contact with the sorbents, soon after it, and at the end after the tests. The oxyhydroxides featured higher performances compared to the oxides in all the cases. In particular, akaganeite had higher As^V^ uptake (89 mg g^−1^ at pH_0_ 3 and 52 mg g^−1^ at pH_0_ 8) when compared with ferrihydrite, both in acidic and basic environments, thanks to the capability to decrease the initial pH, where the surface charge is high and positive. Concerning the As^III^ removal_,_ elevated and steady uptake in the pH_0_ range 2–8 was found for ferrihydrite (≈95% at 100 mg L^−1^, *q_e_* = 144 mg g^−1^ at 500 mg L^−1^ and pH_0_ 8), which was higher than akaganeite (≈80% at 100 mg L^−1^, *q_e_* = 91 mg g^−1^ at 500 mg L^−1^ and pH_0_ 8). The steady behavior in the whole pH range was justified taking into account the presence of the neutral species H_3_AsO_3_, which is not affected by the surface charge of the sorbents, and, therefore, does not diffuse inside the akaganeite nanotubes. Finally, the iron oxide-porous silica composite featured similar performances for As^V^ uptake compared to the bare maghemite, indicating complete accessibility of active sites inside the pores, but dropped down for As^III^ due to the absence of electrostatic interactions between arsenious acid and iron oxide NPs within the pores. Further details on the adsorption of As^V^ on akageneite were obtained by studying the effect of initial concentration, contact time, ionic strength, and presence of competitors. The isotherm plots were best fitted with the Redlich–Peterson model, indicating the presence of energetically equivalent and non-equivalent active sites, especially at pH_0_ 3, where a multilayer may form when the starting concentration exceeds 250 mg L^−1^. The adsorption kinetics at both pH_0_ 3 and 8 was fast and interpreted as pseudo second order, with the equilibrium reached after 120 min. The formation of outer-sphere complexes when electrolytes, such as NaCl and Na_2_SO_4_, are used can cause a slight increase in the removal performances at basic pH_0_ and a decrease at acid ones, higher in the case of sulphate. On the contrary, the formation of inner-sphere complexes in the case of phosphate anions affected the arsenic uptake, ultimately hindering it when present in high concentrations (As:P molar ratio = 1:100).

## Figures and Tables

**Figure 1 nanomaterials-12-00326-f001:**
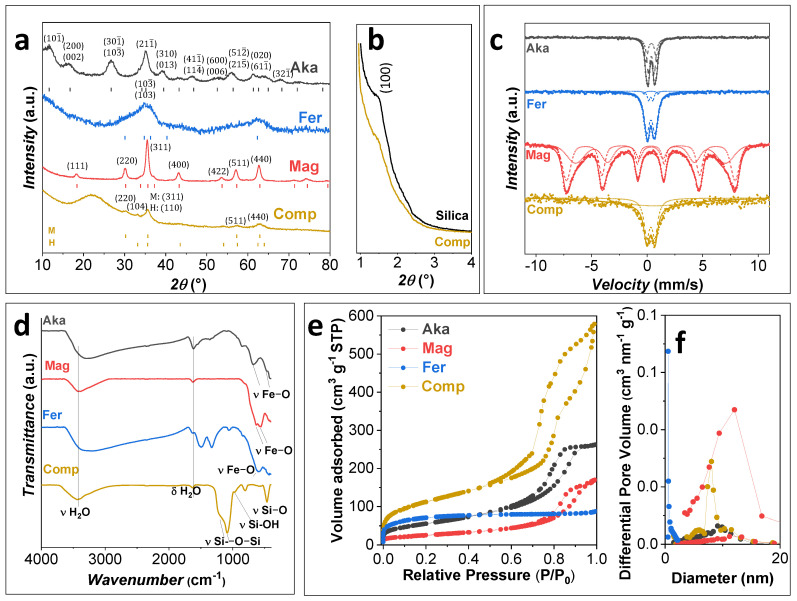
Wide-angle XRD patterns and position of the theoretical XRD diffraction peaks from PDF cards (**a**), small-angle XRD (**b**), ^57^Fe Mössbauer spectra (**c**), FTIR spectra (**d**), N_2_−physisorption isotherms (**e**), and corresponding pore size distribution (**f**) of the samples.

**Figure 2 nanomaterials-12-00326-f002:**
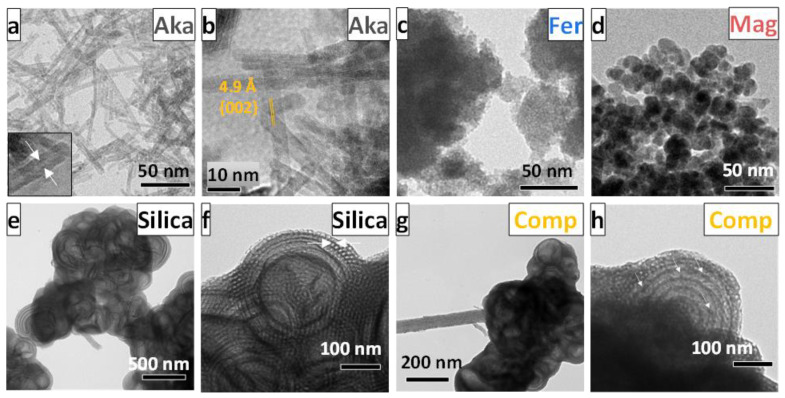
TEM (**a**,**c**–**h**) and HRTEM (**b**) micrographs of Aka, Fer, Mag, silica support (Silica), and corresponding Fe_2_O_3_-silica composite (Comp).

**Figure 3 nanomaterials-12-00326-f003:**
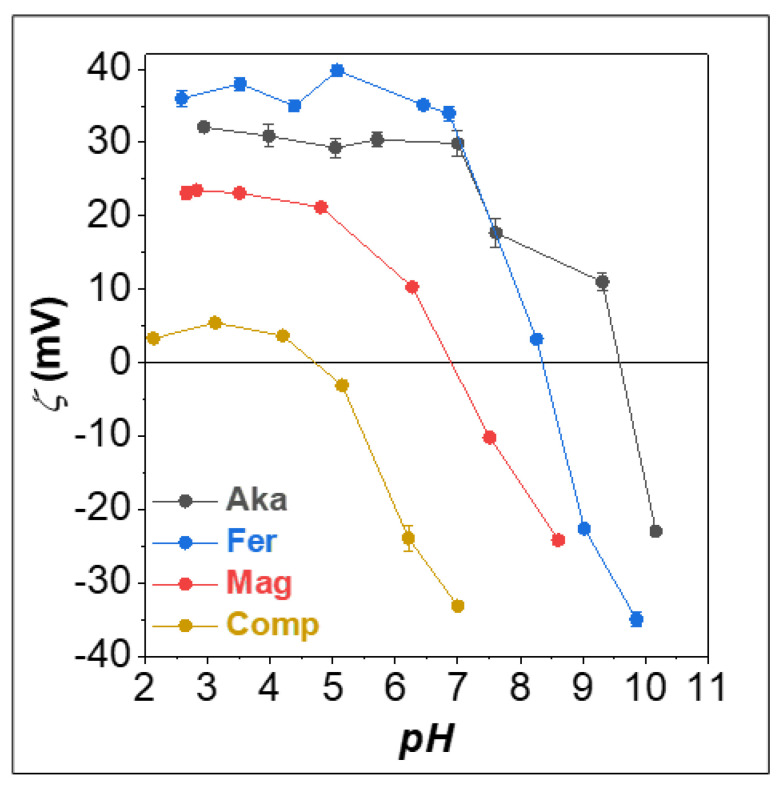
ζ-Potential measurements of the sorbents.

**Figure 4 nanomaterials-12-00326-f004:**
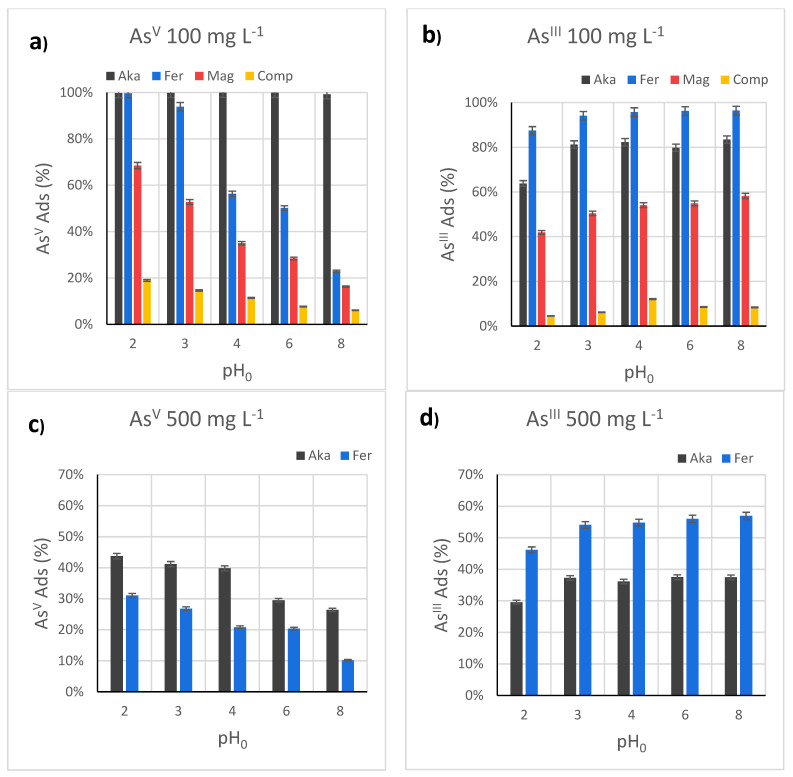
Adsorption capacity from batch adsorption experiments with 100 mg L^−1^ As^V^ (**a**), 500 mg L^−1^ As^V^ (**b**), 100 mg L^−1^ As^III^ (**c**), 500 mg L^−1^ As^III^ (**d**) solution at different initial pH (pH_0_). Aka is expressed in black, Fer in blue, Mag in red, and Comp in orange. Conditions: 20 mL solution, 50 mg of sorbent, 25 °C, sorption time: 16 h. Further details on the adsorption experiments can be seen in Appendix A.

**Figure 5 nanomaterials-12-00326-f005:**
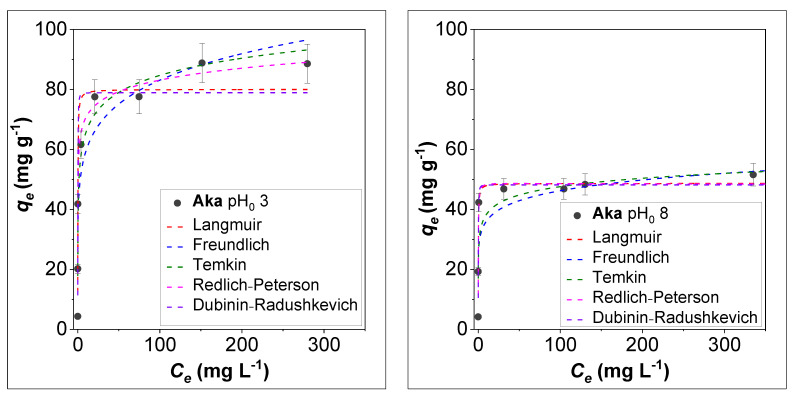
Sorption isotherms of As^V^ on Aka at pH 3 (**left**) and pH 8 (**right**). Conditions: 20 mL As^V^ solution, 50 mg of sorbent dose, adsorption time: 16 h. The isotherms were fitted by Langmuir (red), Freundlich (blue), Temkin (green), Redlich–Peterson (pink), and Dubinin–Radushkevich (violet) model. The corresponding parameters of the adsorption experiments can be seen in Appendix A.

**Figure 6 nanomaterials-12-00326-f006:**
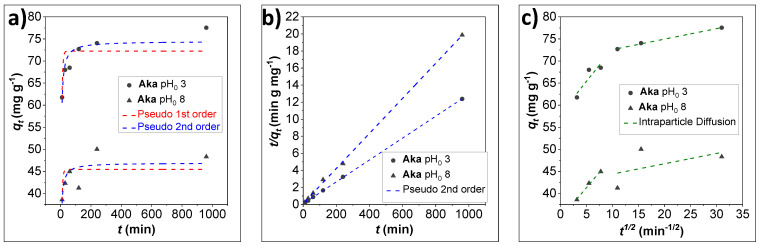
Sorption kinetics of As^V^ on Aka at pH 3 and 8. Kinetics model fitting (**a**), linearized pseudo 2nd order fitting (**b**), and intraparticle diffusion model fitting (**c**). Conditions: 20 mL of 250 mg L^−1^As^V^ solution, 50 mg of sorbent dose, 25 °C, adsorption time: 10–960 min. The corresponding parameters of the adsorption experiments can be seen in Appendix A.

**Figure 7 nanomaterials-12-00326-f007:**
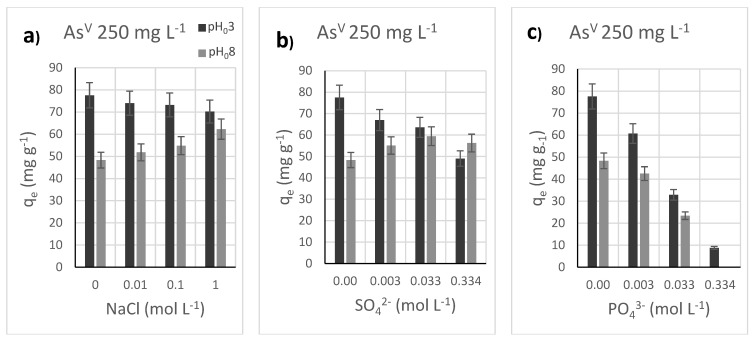
Adsorption capacity from batch adsorption experiments with 250 mg L^−1^ As^V^ solution on Aka at different ionic strengths (**a**), sulphate concentration (**b**), phosphate concentration (**c**), solid/liquid ratio at pH 3 and pH 8. Conditions: 20 mL solution, 50 mg sorbent dose, 25 °C, sorption time: 16 h. Further details on the adsorption experiments can be seen in Appendix A.

**Table 1 nanomaterials-12-00326-t001:** Isotherms models and corresponding parameters.

Model	Equation #	Equation	Parameters	References
Langmuir	Equation (2)	$q_{e}=\frac{q_{m}K_{L}C_{e}}{1+K_{L}C_{e}}$	*q_m_* = maximum adsorption capacity (mg g^−1^) *K_L_* = Langmuir constant (L mg^−1^) *C_e_* = equilibrium concentration (mg L^−1^)	[40]
Frendlich	Equation (3)	$q_{e}=K_{F}C_{e}^{1/n}$	*K_F_* = Freundlich constant (mg^1−1/n^ L^1/n^ g^−1^)	[41]
Temkin	Equation (4)	$q_{e}=\frac{RT}{b_{T}}ln(K_{T}C_{e})$	*R* = universal gas constant (J mol^−1^ K^−1^) *T* = temperature (K) *b_T_* = Temkin parameter (J g mol^−1^ mg^−1^) *K_T_* = Temkin constant (L mg^−1^)	[42]
Redlich–Peterson	Equation (5)	$q_{e}=\frac{K_{RP}C_{e}}{1+\alpha_{RP}C_{e}^{\beta_{RP}}}$	*K_RP_* = Redlich–Peterson constant (L g^−1^) *α_RP_* = Redlich–Peterson parameter 1 (L mg^−1^) *β_RP_* = Redlich–Peterson parameter 2	[43]
Dubinin–Radushkevich	Equation (6)	$q_{e}=q_{m}e^{\left(-K_{DR}\varepsilon_{DR}^{2}\right)}$	*K_DR_* = Dubinin–Radushkevich constant (mol^2^ kJ^−2^) *ε_DR_* = Dubinin–Radushkevich variable (kJ mol^−1^)	[44]

**Table 2 nanomaterials-12-00326-t002:** Kinetics models and corresponding parameters.

Model	Equation #	Equation	Parameters
Pseudo 1st-Order	Equation (9)	$q_{t}=q_{e1}\left(1-e^{\left(K^{\prime }t\right)}\right)$	*K*′ = pseudo-1st order constant (min^−1^)
Pseudo 2nd-Order	Equation (10)	$q_{t}=\frac{K^{\prime \prime }q_{e2}^{2}t}{1+K^{\prime \prime }q_{e2}t}$	*K*″ = pseudo-2nd order constant (g mg^−1^ min^−1^)
Intraparticle diffusion model	Equation (12)	$q_{t}=k_{i}t^{\frac{1}{2}}+x_{i}$	*k_i_* = intraparticle diffusion constant (mg g^−1^ min^−1/2^)

**Table 3 nanomaterials-12-00326-t003:** Structural parameters of the sorbents extracted from the Rietveld refinement of XRD patterns. In the case of anisotropic Aka, the anisotropic-no-rules model was employed. Morphological parameters calculated from TEM micrographs. Textural parameters calculated from N_2_-physisorption experiments. V_P_ for the Fer sample was calculated by the Horvat Kawazoe model, while a BJH model was adopted for the other samples.

Sample	Phase	a (Å)	b (Å)	c (Å)	D_XRD_ ^1^ (nm)	D_XRD_ ^2^ (nm)	D_TEM_ ^1^ (nm)	D_TEM_ ^2^ (nm)	S_BET_ (m^2^ g^−1^)	V_p_ (cm^3^ g^−1^)	D_p_ (nm)
Aka	Isotropic akaganeite	10.57 (1)	3.030 (1)	10.48 (1)	5.3 (1)	-	57 (16)	4.3 (8)	202 (4)	0.327 (3)	9.4 (2)
	Anisotropic akaganeite				2.0 (6)	25.1 (2)					
Mag	Maghemite	8.379 (1)	=a	=a	14.0 (1)	n.a.	12 (3)	n.a.	92 (2)	0.156 (2)	11.8 (2)
Fer	Ferrihydrite	5.69 (5)	=a	9.03 (9)	1.7 (3)	n.a.	4 (1)	n.a.	260 (5)	0.110 (2)	0.73 (1)
Comp	Hematite 18(2)%	5.052 (6)	=a	13.74 (2)	9.1 (6)	n.a.	9 (2)	n.a.	410 (9)	0.594 (6)	8.1 (2)
	Maghemite 82(8)%	8.357 (6)	=a	=a	6.6 (2)	n.a.		n.a.			
Silica	n.a.	n.a.	n.a.	n.a.	n.a.	n.a.	n.a.	n.a.	457 (9)	0.666 (7)	7.7 (2)

a, b, and c: cell parameters; D_XRD_
^1^ and D_XRD_
^2^: crystallite sizes; D_TEM_
^1^ and D_TEM_
^2^: particle sizes; S_BET_: surface area; V_p_: pore volume; D_p_: pore diameter.

**Table 4 nanomaterials-12-00326-t004:** Isotherm fitting parameters for adsorption of As^V^ onto Aka at pH 3 and 8.

Sample	pH_0_	Isotherm	R^2^	K	q_m_ (mg g^−1^)	n	b_T_ (kJ g mol^−1^ mg^−1^)	α_RP_ (L mg^−1^)	β_RP_	E_ads_ (kJ mol^−1^)
Aka	3	L	0.93	6 (2)	80 (4)	-	-	-	-	-
		F	0.88	42 (6)	-	0.15 (3)	-	-	-	-
		T	0.95	3 (2)∙10^2^	-	-	0.30 (3)	-	-	-
		RP	0.97	7 (3)∙10^2^	-	-	-	12 (6)	0.93 (3)	-
		DR	0.92	2.5 (6) 10^−2^	79 (4)	-	-	-	-	4(1)
	8	L	0.91	6 (2)	49 (2)	-	-	-	-	-
		F	0.68	28 (5)	-	0.11 (4)	-	-	-	-
		T	0.75	1 (3)∙10^3^	-	-	0.6 (1)	-	-	-
		RP	0.92	3 (1)∙10^2^	-	-	-	6 (3)	1.00 (4)	-
		DR	0.91	2.7 (6) 10^−2^	48 (2)	-	-	-	-	4(1)

L = Langmuir; F = Freundlich; T = Temkin; RP = Redlich–Peterson; DR = Dubinin–Radushkevich. * Constants units: K_L_ (L mg^−1^); K_F_ (mg^1−1/n^ L^1/n^ g^−1^); K_T_ (L mg^−1^); K_RP_ (L g^−1^); K_DB_ (mg L^−1^).

## Data Availability

Data is contained within the article or Appendix A.

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
