# Peer review of "As(III, V) Uptake from Nanostructured Iron Oxides and Oxyhydroxides: The Complex Interplay between Sorbent Surface Chemistry and Arsenic Equilibria"

_nanomaterials, 2022, doi:10.3390/nano12030326_

Round 1
Reviewer 1 Report
It is a well-executed and well-written article possessing significant novelty and importance to the environmental field. The results have been richly modelled and interpreted with appropriate scientific content. Therefore, I recommend acceptance of this research article after addressing the following minor issues:
- There are too many references (120) for this research article. It should be reduced be around 45. Probably, the authors may consider writing a review article using most of the cited articles. But, in this article, it doesn’t need that many references.
- In the section 2.1 and those sections proving the name of instruments used, the city and country names should be provided alongside the company from where it was procured. In the case of USA, the details of city, state and country should be provided.
- A reference citation should be added for the preparation methods of maghemite and silica-iron oxide composite.
- All the units and description of terms in Eq.1-12 should be double-checked for correctness.
- All the short paragraphs in Section 2.6 should be combined to be one paragraph.
- The peaks in Fig.1a,b,c,e,f should be labelled as shown for Fig.1d.
- All the abbreviations should be provided in the full form under the Table 1 as footnote.
- The sub-heading 3.2 should be removed. Instead all the sub-sections under section 3.2 should be numbered as 3.2, 3.3, 3.4, 3.5 and so on. Also, several topics under section 3.2.2 should be provided with separate sub-headings like “effect of initial concentration and isotherm modelling”, “effect of contact time and kinetic modelling”, “effect of added salts as competitors” and so on.
- Table 7S should be brought to the main text.
- Figures 4 and 7 – the x-axis and y-axis lines should be drawn all the plots. Also, no need of decimal point percentage values in Fig.4 and qe values in Fig.7.
Author Response
Dear Editor and Reviewers,
thank you for the suggestions and comments to improve the quality of our manuscript.
Please, see below a point-by-point answer.
Referee 1
It is a well-executed and well-written article possessing significant novelty and importance to the environmental field. The results have been richly modelled and interpreted with appropriate scientific content. Therefore, I recommend acceptance of this research article after addressing the following minor issues:
- There are too many references (120) for this research article. It should be reduced be around 45. Probably, the authors may consider writing a review article using most of the cited articles. But, in this article, it doesn’t need that many references.
Answer: Taking into account the suggestion of the second reviewer of adding 3 references and the citation for the preparation of maghemite and silica composite, we had been able to reach a final number of 68 references, removing more than 50 papers. We hope that this can be considered a good compromise, since also in other recently published Nanomaterials articles, this is the average number of citations.
- In the section 2.1 and those sections proving the name of instruments used, the city and country names should be provided alongside the company from where it was procured. In the case of USA, the details of city, state and country should be provided.
Answer: We have added the information on the instruments used in section 2.6.
- A reference citation should be added for the preparation methods of maghemite and silica-iron oxide composite.
Answer: The references for the synthesis methods have been added.
- All the units and description of terms in Eq.1-12 should be double-checked for correctness.
Answer: All units and equations have been checked and corrected as highlighted in the text.
- All the short paragraphs in Section 2.6 should be combined to be one paragraph.
Answer: Paragraph 2.6 was revised accordingly.
- The peaks in Fig.1a,b,c,e,f should be labelled as shown for Fig.1d.
Answer: We have added the Miller’s indexes (Fig. 1a,b). For the figures 1c,e,f the peaks have different meanings that cannot be simply depicted by labels.
- All the abbreviations should be provided in the full form under the Table 1 as footnote.
Answer: The table footnote has been added accordingly.
Table 1. Structural parameters of the sorbents extracted from the Rietveld refinement of XRD patterns. In the case of anisotropic Aka, the anisotropic-no-rules model was employed. Morphological parameters calculated from TEM micrographs. Textural parameters calculated from N2-physisorption experiments. VP for the Fer sample was calculated by the Horvat Kawazoe model, while a BJH model was adopted for the other samples.
|
Sample |
Phase |
a (Å) |
b (Å) |
c (Å) |
DXRD1 (nm) |
DXRD2 (nm) |
DTEM1 (nm) |
DTEM2 (nm) |
SBET (m2 g-1) |
Vp (cm3 g-1) |
Dp (nm) |
|
Aka |
Isotropic akaganeite |
10.57(1) |
3.030(1) |
10.48(1) |
5.3(1) |
- |
57(16) |
4.3(8) |
202(4) |
0.327(3) |
9.4(2) |
|
Anisotropic akaganeite |
2.0(6) |
25.1(2) |
|||||||||
|
Mag |
Maghemite |
8.379(1) |
=a |
=a |
14.0(1) |
n.a. |
12(3) |
n.a. |
92(2) |
0.156(2) |
11.8(2) |
|
Fer |
Ferrihydrite |
5.69(5) |
=a |
9.03(9) |
1.7(3) |
n.a. |
4(1) |
n.a. |
260(5) |
0.110(2) |
0.73(1) |
|
Comp |
Hematite 18(2)% |
5.052(6) |
=a |
13.74(2) |
9.1(6) |
n.a. |
9(2) |
n.a. |
410(9) |
0.594(6) |
8.1(2) |
|
Maghemite 82(8)% |
8.357(6) |
=a |
=a |
6.6(2) |
n.a. |
n.a. |
|||||
|
Silica |
n.a. |
n.a. |
n.a. |
n.a. |
n.a. |
n.a. |
n.a. |
n.a. |
457(9) |
0.666(7) |
7.7(2) |
a, b, and c: cell parameters; DXRD1 and DXRD2: crystallite sizes; DTEM1 and DTEM2: particle sizes; SBET: surface area; Vp: pore volume; Dp: pore diameter.
- The sub-heading 3.2 should be removed. Instead all the sub-sections under section 3.2 should be numbered as 3.2, 3.3, 3.4, 3.5 and so on. Also, several topics under section 3.2.2 should be provided with separate sub-headings like “effect of initial concentration and isotherm modelling”, “effect of contact time and kinetic modelling”, “effect of added salts as competitors” and so on.
Answer: We have revised the titles of the paragraphs as suggested.
- Table 7S should be brought to the main text.
Answer: The Table has been added in the main manuscript as Table 4.
Table 2 Isotherm fitting parameters for adsorption of AsV onto Aka at pH 3 and 8.
|
Sample |
pH0 |
Isotherm |
R2 |
K |
qm (mg g-1) |
n |
bT (kJ g mol-1 mg-1) |
αRP (L mg-1) |
βRP |
Eads (kJ mol-1) |
|
Aka |
3 |
L |
0.93 |
6(2) |
80(4) |
- |
- |
- |
- |
- |
|
F |
0.88 |
42(6) |
- |
0.15(3) |
- |
- |
- |
- |
||
|
T |
0.95 |
3(2)∙102 |
- |
- |
0.30(3) |
- |
- |
- |
||
|
RP |
0.97 |
7(3)∙102 |
- |
- |
- |
12(6) |
0.93(3) |
- |
||
|
DR |
0.92 |
2.5(6) ∙10-2 |
79(4) |
- |
- |
- |
- |
4(1) |
||
|
8 |
L |
0.91 |
6(2) |
49(2) |
- |
- |
- |
- |
- |
|
|
F |
0.68 |
28(5) |
- |
0.11(4) |
- |
- |
- |
- |
||
|
T |
0.75 |
1(3)∙103 |
- |
- |
0.6(1) |
- |
- |
- |
||
|
RP |
0.92 |
3(1)∙102 |
- |
- |
- |
6(3) |
1.00(4) |
- |
||
|
DR |
0.91 |
2.7(6) ∙10-2 |
48(2) |
- |
- |
- |
- |
4(1) |
L = Langmuir; F = Freundlich; T = Temkin; RP = Redlich-Peterson; DR = Dubinin-Radushkevich. *Constants units: KL (L mg-1); KF (mg1−1/n L1/n g−1); KT (L mg‑1); KRP (L g-1); KDB (mg L-1)
- Figures 4 and 7 – the x-axis and y-axis lines should be drawn all the plots. Also, no need of decimal point percentage values in Fig.4 and qe values in Fig.7.
Answer: The figures 4 and 7 has been revised to address these issues.
Referee 2
This well-written, complete and rigorous article proposes a study of the arsenic retention by iron
oxides nanoparticles.
For my part, this article could be published after making minor changes: citations should be
written before the punctuation mark, not after. Please, separate the units of the number they
accompany. Also, hunt double spaces between words.
Answer: The citations and the units have been modified as suggested. Moreover, the manuscript has been checked for double spaces and corrected.
More particular comments:
- INTRODUCTION
The first paragraph is too long. Please separate it in two.
Answer: the first paragraph has been divided in two.
Please, add some references to nanoparticles (line 49)
(https://pubs.rsc.org/en/content/articlelanding/2014/ay/c4ay02181a/unauth;
https://www.sciencedirect.com/science/article/pii/S0304389420320379?casa_token=7CwmUjZfF_wAAAAA:-xvmp2Es0u_nqRubdjJj7Ya2Hd8o3Huz84QTLldG8IXo2q_i0RgtN9zDHvUV9LcxEXAZwspC1A).
In this way, the importance of nanoparticles in the retention of arsenic is highlighted. Also, take a look at this newly published review
(https://www.sciencedirect.com/science/article/pii/S0304389421025401?casa_token=HkCIIkgG1oAAAAA:8C_vl9ldkWGEWsK72np2XSiVbVJ27ehOdZppxWpK272WvbkIM5dyDN3kBsgXvQ6P4buuQVNQg).
Answer: The references have been all cited in the text.
- MATERIALS AND METHODS
What was the precursor compound for arsenic(V)?
Answer: As reported in line 135, the precursor compound was Na2HAsO4·7H2O.
Line 88. Change Merk for Merck.
Answer: The typo has been corrected.
Sections 2.4 and 2.5. It might be interesting to include both equations and explanation of the
parameters in two tables, one for each section.
Answer: The equations have been summarized in tables 1 and 2.
Table 3. Isotherms models and corresponding parameters.
|
Model |
Eq. # |
Equation |
Parameters |
Reference |
|
Langmuir |
Eq. 2 |
qm = maximum adsorption capacity (mg g-1) KL = Langmuir constant (L mg-1) Ce = equilibrium concentration (mg L-1) |
[40] |
|
|
Frendlich |
Eq. 3 |
KF = Freundlich constant (mg1−1/n L1/n g−1) |
[41] |
|
|
Temkin |
Eq. 4 |
R = universal gas constant (J mol-1 K-1) T = temperature (K) bT = Temkin parameter (J g mol-1 mg-1) KT = Temkin constant (L mg-1) |
[42] |
|
|
Redlich-Peterson |
Eq. 5 |
KRP = Redlich-Peterson constant (L g-1) αRP = Redlich-Peterson parameter 1 (L mg-1) βRP = Redlich-Peterson parameter 2 |
[43] |
|
|
Dubinin-Radushkevich |
Eq. 6 |
KDR = Dubinin-Radushkevich constant (mol2 kJ-2) εDR = Dubinin-Radushkevich variable (kJ mol-1) |
[44] |
Table 4. Kinetics models and corresponding parameters.
|
Model |
Eq. # |
Equation |
Parameters |
|
Pseudo 1st-Order |
Eq. 9 |
K' = pseudo-1st order constant (min-1) |
|
|
Pseudo 2nd-Order |
Eq. 10 |
K" = pseudo-2nd order constant (g mg-1 min-1) |
|
|
Intraparticle diffusion model |
Eq. 12 |
ki = intraparticle diffusion constant (mg g-1 min-1/2) |
- RESULTS AND DISCUSSION
Please, put the figures’ references in other colour.
Answer: The references have been turned into black.
Page 9, try to fill in the gap. You could incorporate some of the text that is after Figure 4.
Answer: The text has been revised accordingly.
Figure 4 has two figures 4.c. Please, correct. In addition, I think the bar chart is not the best type
of graphic to can use in this case. The authors should try the points joined by a line, through which trends would be better appreciated. Try it in Figure 7 too.
Answer: The label of figure 4b has been corrected. We have tried to follow the referee’s suggestion concerning the use of line and points instead of a bar chart. As can be seen in the following figure, in our opinion the bar chart is clearer to the reader. Nevertheless, we are willing to change the figures if the reviewers and editor will prefer the new version.

Reviewer 2 Report
I make my comments in the attached document.

Author Response

(The authors gave the same response as above.)
